# Comparison of risks of arterial thromboembolic events and glaucoma with ranibizumab and aflibercept intravitreous injection: A nationwide population-based cohort study

Yin-Hsi Chang[1,2☯], Li-Nien Chien[3,4☯], Wan-Ting Chen[3], I-Chan Lin[5,6]*

1 Department of Ophthalmology, Chang Gung Memorial Hospital, Linkou Medical Center, Taoyuan, Taiwan,
2 School of Medicine, College of Medicine, Taipei Medical University, Taipei, Taiwan, 3 School of Health Care Administration, College of Management, Taipei Medical University, Taipei, Taiwan, 4 Health Data Analytics and Statistics Center, Office of Data Science, Taipei Medical University, Taipei, Taiwan, 5 Department of Ophthalmology, Taipei Medical University, Shuang Ho Hospital, New Taipei, Taiwan, 6 Department of Ophthalmology, School of Medicine, College of Medicine, Taipei Medical University, Taipei, Taiwan

☯ These authors contributed equally to this work.
* ichanlin@gmail.com

**Data Availability Statement:** Data are available from the Health and Welfare Science Data Center (HWDC), Ministry of Health and Welfare in Taiwan

## Abstract

### Background

To compare intravitreal aflibercept injection with intravitreal ranibizumab injection for the risk of major arterial thromboembolic events (ATEs) and glaucoma.

### Methods

This retrospective, nationwide cohort study investigated 15 611 and 3867 patients aged >50 years with at least one pharmacy claim for intravitreal ranibizumab injection and aflibercept injection between 2011 and 2016, respectively. The inverse probability of treatment weighting method was performed to adjust the baseline difference between the two groups and the hazard risk of adverse events was estimated using the Cox proportional regression model.

### Results

No significant difference was noted between intravitreal ranibizumab and aflibercept injection for arterial thromboembolic risk, including ischemic stroke and acute myocardial infarction, during a 2-year follow-up (adjusted hazard ratio (HR): 0.87, 95% confidence interval (CI): 0.53–1.42; $P$ = .583). Subgroup analyses revealed that patients age >65 years (adjusted HR: 0.64, 95% CI: 0.45–0.92) and those without coronary artery disease (adjusted HR: 0.59, 95% CI: 0.37–0.95) had significantly lower arterial thromboembolic risk in the aflibercept group than in the ranibizumab group. Additionally, the risk of glaucoma development after intravitreal injection did not significantly differ between the two groups (adjusted HR: 0.63, 95% CI: 0.37–1.06; $P$ = .084).

(http://dep.mohw.gov.tw/DOS/np-2497-113.html). Due to legal restrictions imposed by the government of Taiwan in relation to the Personal Information Protection Act, data cannot be made publicly available. Contact information for data application, analysis, and inquiry (https://dep.mohw.gov.tw/dos/cp-2516-59203-113.html).

**Funding:** The authors received no specific funding for this work.

**Competing interests:** The authors have declared that no competing interests exist.

## Conclusions

No significant differences in the risk of major ATEs and glaucoma were found between ranibizumab and aflibercept, and aflibercept might be safe for use in elderly patients.

## Introduction

Antivascular endothelial growth factor (anti-VEGF) agents play a major role in the treatment of many retinal diseases because they inhibit VEGF angiogenic activity and prevent neovascularization [1–3]. Numerous anti-VEGF agents, including pegaptanib, bevacizumab, ranibizumab, and aflibercept, have been used in clinical practice [4]. Ranibizumab (Lucentis, Genentech, Inc., South San Francisco, CA, USA) is a recombinant, humanized, monoclonal, VEGF-specific antibody fragment that inhibits all VEGF-A isoforms. Ranibizumab as a treatment for retinal neovascularization diseases such as neovascular age-related macular degeneration (nAMD) was approved for use by the United States Food and Drug administration (FDA) in June 2006 and by the Taiwan FDA since 2009. Aflibercept (Eylea, Regeneron, Tarrytown, PA, USA and Bayer HealthCare, Berlin, Germany) was also approved by the US FDA in November 2011 and by the Taiwan FDA in June 2013 for treating nAMD, and its use has rapidly increased since then [5]. Aflibercept is a humanized fusion protein that binds VEGF-A, VEGF-B, and placental growth factor with stronger binding affinity than that of ranibizumab [6]. In Taiwan, currently, these two agents are the treatment of choice for nAMD, diabetic macular edema (DME), central and branch retinal vein occlusion (RVO), myopic choroidal neovascularization (mCNV), and polypoidal choroidal vasculopathy and are considered comparably effective [7, 8].

Nevertheless, side effects related to the use of these agents have been reported. Because VEGF stimulates nitric oxide production, vasodilation, and antithromboticity, the anti-VEGF activity is associated with thrombogenicity [9]. This raises concerns regarding systemic adverse effects, such as acute myocardial infarction (AMI) and ischemic stroke, because the drugs potentially enter the circulation through uveal vessels or through aqueous humor outflow [10, 11]. In patients receiving intravitreal ranibizumab (IVR), the overall arterial thromboembolic event (ATE) rate was approximately 2.2%–6.6% depending on the dosage [10]. Conversely, the ATE incidence rate was 2.19 per 100 person-years in patients who received 2 mg of intravitreal aflibercept (IVA) [11]. The Diabetic Retinopathy Clinical Research Network (DRCR.net) conducted a comparative effectiveness trial comparing aflibercept, bevacizumab, and ranibizumab in the treatment of DME associated with visual impairment [2]. All three regimens produced substantial visual acuity (VA) improvement through 2 years. However, systemic Anti-Platelet Trialists' Collaboration (APTC) rates were higher in the ranibizumab group, with a greater number of nonfatal strokes and vascular deaths in the ranibizumab group. However, these findings are inconsistent with other studies. Several studies have reported the systemic risks of acute MI and acute cerebrovascular disease (CVD) were not significantly higher among patients treated with these three agents [12, 13]. Given this inconsistency, whether the risk of APTC events is higher with ranibizumab use than with aflibercept or bevacizumab use remains uncertain.

Anti-VEGF agents are administered through intravitreal injection (IVI). Intraocular pressure (IOP) usually rapidly increases and then decreases within 1 hour after the injection, but some reports have indicated that multiple injections may lead to a long-term increase in IOP [14]. Few studies have compared the risk of elevated IOP between ranibizumab and aflibercept

intravitreal injection [15–17]. Because IOP elevation may increase the risk of glaucoma development, which affects vision, the association between IVI and glaucoma should be evaluated.

In this study, we compared the rates of a systemic adverse event, ATE, and an ocular adverse event, glaucoma, between patients receiving IVR and those receiving IVA by using a nationally representative sample in Taiwan to elucidate the real-word condition.

## Materials and methods

### Institutional review board and data set

This study obtained data from the National Health Insurance Research Database (NHIRD) provided by the Health and Welfare Data Science Center (HWDC), Ministry of Health and Welfare (MOHW), Taiwan. The NHIRD is managed by HWDC, and the data are released for research purposes only, with confidentiality being maintained according to the directives of the National Health Insurance Administration (NHIA). The NHIRD is a reimbursement claims database that covers 99% of the residents in Taiwan enrolled in the National Health Insurance (NHI) program [18]. The NHI program is a single-payer health insurance system that has a contract with most health-care providers in Taiwan. NHI provides a universal coverage for all necessary medical expenses including outpatient visits, inpatient systems, prescriptions, traditional Chinese medicine, dental services, operations, and investigations such as X-rays or magnetic resonance imaging. Disease diagnoses were coded using the *International Classification of Diseases*, *9th Revision*, *Clinical Modification (ICD-9-CM)* and since 2016, they are coded using the *International Classification of Diseases*, *10th revision*, *Clinical Modification (ICD-10-CM)* codes [19]. Care providers must upload the claims data from each service to the NHIA. In addition, care providers must upload the medical records of reimbursement applications for ranibizumab and aflibercept. NHI approval for ranibizumab and aflibercept reimbursement is not only based on medical coding, but also includes a review of the applicant's medical records by retinal specialists.

In this study, we also obtained patient death records from the National Death Registry. The National Death Registry records the deaths of all citizens; causes of death are coded using information on death certificates. The accuracy of the coding has been validated by previous studies [20, 21]. The NHIRD and the National Death Registry can be linked by a unique encrypted ID. Because of privacy issues, this data linkage can only be processed by researchers at the HWDC. The data were de-identified, so researchers were unable to obtain the informed consents from the study participants. This Joint Institutional Review Board of Taipei Medical University approved the study protocol (TMU-JIRB No.202005104).

### Study cohort

The study cohort comprised patients who received IVR or IVA between 2011 and 2016. This study period was chosen because ranibizumab and aflibercept were first reimbursed by NHI in Taiwan on January 1, 2011, and August 1, 2014, respectively. Data of outcomes were collected until December 31, 2017, to ensure that at least 1 year of follow-up data were available for all eligible patients. The first prescription claim for IVR or IVA was treated as the index date. NHI covers IVR and IVA in the treatment of nAMD, DME, central RVO, and mCNV. We excluded patients (1) aged <50 years, (2) with missing gender information, (3) who had received IVI before the index date in order to exclude patients who had self-paid for medication for any indication, and (4) who received both ranibizumab and aflibercept simultaneously during the study period. The detailed flowchart of patient selection is presented in Fig 1.

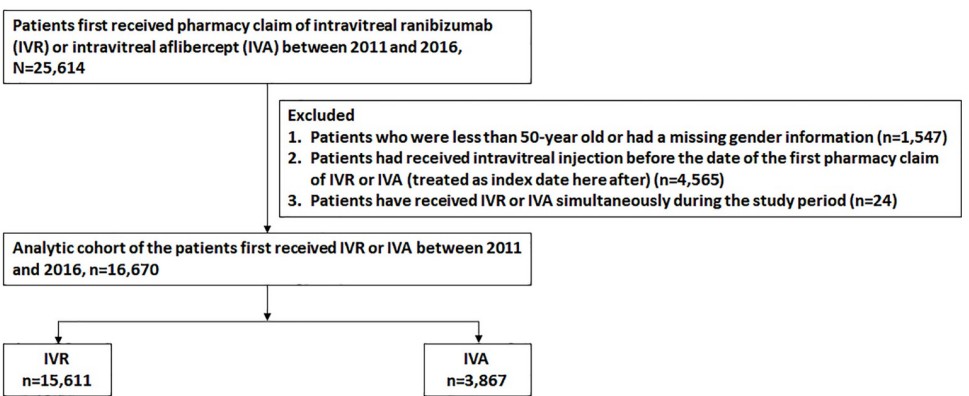

**Fig 1. Flowchart of patient selection.** IVA = intravitreal aflibercept; IVR = intravitreal ranibizumab.

## Inverse probability treatment weighting

To reduce the potential selection bias, we used inverse probability treatment weighting (IPTW) based on the propensity score to balance the baseline characteristics between patients receiving ranibizumab and aflibercept to ensure the two groups have similar distributions of observed baseline covariates [22]. The IPTW approach could be applied the entire cohort and had the advantage of addressing a large number of confounding variables instead of matching two treatment individuals based on a selected group of confounders. We considered the covariates of age, sex, CHA2DS2-VASc score, comorbidities, and medications to estimate the weight (Table 1). The CHA2DS2-VASc score is the most commonly used measure for predicting thromboembolic risk in atrial fibrillation. CHA2DS2 stands for (**C**ongestive heart failure, **H**ypertension, **A**ge (50–64 = 0 point, 65–74 = 1 point, ≥75 = 2 points), **D**iabetes, and previous **S**troke/transient ischemic attack (2 points). **VASc** stands for vascular disease (peripheral arterial disease, previous myocardial infarction, aortic atheroma), and sex category (female sex) is also included in this scoring system [23]. Each individual in the cohort was assigned a weight based on the likelihood of exposure to the treatment effect under investigation. The CHA2DS2-VASc score was applied to adjust the stroke risk between the two study groups. Underlying vascular comorbidities, including hypertension, diabetes, renal disease, atrial fibrillation, and coronary artery disease (CAD), were defined as more than two diagnostic claims made one year before the index date of IVR or IVA treatment. The *ICD-9-CM* and *ICD-10-CM* codes for disease diagnosis and the Anatomical Therapeutic Chemical codes for medications are listed in S1 Table.

## Main outcome measurements

The primary outcome measurements were the association between intravitreal therapy and major ATEs, including AMI and ischemic stroke, as well as the association between intravitreal therapy and glaucoma. An AMI event was identified based on *ICD-9-CM* or *ICD-10-CM* diagnostic codes with hospitalization and antiplatelet administration. An ischemic stroke event was identified based on *ICD-9-CM* or *ICD-10-CM* diagnostic codes with hospitalization and brain imaging (either computed tomography or magnetic resonance imaging) after IVI.

Glaucoma was defined as at least three times of diagnosis based on *ICD-9-CM* or *ICD-10-CM* diagnostic codes with antiglaucoma agents (ATC code S01E) administration. The *ICD-9-CM* and *ICD-10-CM* codes for glaucoma diagnosis are listed in S1 Table. 3 recorded visits

**Table 1. Baseline characteristics of patients who received IVR or IVA before and After IPTW.**

| | Before IPTW | | | After IPTW | | |
|---|---|---|---|---|---|---|
| | IVR | IVA | SMD | IVR | IVA | SMD |
| | (n = 15 611) | (n = 3867) | | (n = 15 611) | (n = 3867) | |
| **Year of first injection** | | | | | | |
| 2011–2012 | 12.5 | 0.0 | 0.534 | 14.3 | 0.0 | 0.579 |
| 2013–2014 | 43.2 | 7.8 | 0.889 | 43.1 | 6.5 | 0.937 |
| 2015–2016 | 44.3 | 92.2 | 1.200 | 42.5 | 93.5 | 1.305 |
| **Male** | 59.3 | 62.1 | 0.059 | 59.9 | 61.7 | 0.036 |
| **Age (y), mean (SD)** | 68.1 ± 10.0 | 72.0 ± 10.0 | 0.389 | 68.1±10.1 | 70.8 ± 10.0 | 0.113 |
| **Age group** | | | | | | |
| 50–64 | 40.8 | 24.5 | 0.353 | 37.3 | 33.4 | 0.083 |
| 65–74 | 32.1 | 33.2 | 0.024 | 32.3 | 32.7 | 0.008 |
| 75– | 27.1 | 42.3 | 0.323 | 30.3 | 33.9 | 0.077 |
| **Diseases diagnosis** | | | | | | |
| nAMD | 45.5 | 52.7 | 0.144 | 46.7 | 45.0 | 0.036 |
| Diabetic maculopathy | 34.8 | 6.7 | 0.740 | 29.2 | 28.7 | 0.010 |
| RVO | 1.2 | 0.7 | 0.046 | 1.1 | 1.5 | 0.037 |
| Others | 18.5 | 40.0 | 0.485 | 23.0 | 24.8 | 0.042 |
| **$CHA_2DS_2$-VASc** | | | | | | |
| Mean (SD) | 2.8 ± 1.4 | 2.6 ± 1.5 | 0.134 | 2.8 ± 1.4 | 2.8 ± 1.5 | 0.012 |
| 0–1 | 17.0 | 25.4 | 0.207 | 18.8 | 18.4 | 0.009 |
| ≥2 | 83.0 | 74.6 | 0.207 | 81.2 | 81.6 | 0.009 |
| **Comorbidity, yes** | | | | | | |
| Hypertension | 62.5 | 54.8 | 0.155 | 60.9 | 59.9 | 0.020 |
| Diabetes | 63.3 | 27.7 | 0.765 | 56.0 | 53.8 | 0.045 |
| Renal disease | 12.5 | 6.9 | 0.193 | 11.4 | 10.6 | 0.024 |
| Atrial fibrillation | 2.1 | 2.6 | 0.033 | 2.2 | 2.7 | 0.033 |
| CAD | 16.4 | 15.7 | 0.018 | 16.4 | 18.7 | 0.063 |
| IS or AMI hospitalization | 6.2 | 4.3 | 0.085 | 5.8 | 5.1 | 0.030 |

Values are % or mean ± SD.

nAMD = neovascular age-related macular degeneration; AMI = acute myocardial infraction; CAD = coronary artery disease; $CHA_2DS_2$-VASc score = congestive heart failure, hypertension, age ≥ 75 years, diabetes, stroke/transient ischemic attack, vascular disease, age 65–74 years, sex category (female); RVO = retinal vein occlusion; IPTW = inverse probability of treatment weighting; IS = ischemic stroke; IVA = intravitreal aflibercept injection; IVR = intravitreal ranibizumab injection; SMD = standardized mean difference

should be with the same diagnosis code, and the 3 visits should be at least 28 days part. These cases of glaucoma were all diagnosed by certified ophthalmologists Patients with any recoded visits for a diagnosis of glaucoma or those have been prescribed antiglaucoma agents before receiving the first intravitreal injection were excluded.

## Sensitivity and subgroup analyses

Sensitivity analyses were performed by varying the history of ATEs and follow-up length from 1 to 2 years. Subgroup analyses were performed to evaluate the risks of ATEs and the baseline characteristics among the patients, including age, sex, medical comorbidities, CHA2DS2--VASc score, retinal disease, and the history of ATEs.

## Statistical analysis

A logistic regression was used to calculate propensity scores. The standardized mean difference (SMD) was used to compare the baseline characteristics between the two groups, and an SMD <10% (or 0.1) indicated negligible correlation between the variables of the treatment groups [24]. The primary analyses were based on the incidence computed as the number of events per 100 person-years (PY) because the number of patients in each year varied between the IVR and IVA groups and the PY methodology accounted for both the number of patients at risk and the exposure duration at the time of the risk. The follow-up period was set from the index date to whichever of the following occurred first: (1) ischemic stroke or AMI hospitalization, (2) death, (3) switch to another intravitreal medication, or (4) December 31, 2017. Our study design ensured a follow-up of at least 1 years.

The adjusted hazard ratio (HR) was calculated using the Cox proportional hazard model adjusted for sex, age, comorbidities, and prescribed medications and 95% confidence intervals (CIs) were calculated separately for each analysis. The IVR group was set as a reference, and a CI containing 1.00 implied no statistical difference between the treatment groups. The cumulative event rates of interest were estimated based on the Kaplan–Meier method. A stratified Cox proportional hazard regression was used to compare the risk of events between the IVR group and IVA group. The assumption of proportional hazards was assessed. All analyses were performed using SAS/STAT 9.4 (SAS Institute Inc., Cary, NC, USA) and STATA 14 (Stata Corp LP, College Station, TX, USA). A value of $P < .05$ was considered significant.

## Results

In total, 25614 patients between 2011 and 2016 were screened for study eligibility, and 19478 (76%) were included in the study cohort (Fig 1). In the cohort, 15611 (80%) patients received ranibizumab, and 3867 (20%) received aflibercept injection. No significant difference was observed between the two groups in baseline characteristics and after IPTW in terms of age, sex, diagnosis, cardiovascular risk, and comorbidities (Table 1). The mean follow-up periods of the ranibizumab and aflibercept groups were 1.9 (±0.3) and 1.6 (±0.4) years, respectively.

For the risks of ATEs, no significant difference was observed between the IVA and IVR groups in our study cohort in the 2-year follow-up period after sensitivity analysis (adjusted HR: 0.86, 95% CI: 0.57–1.32; $P = .498$; Fig 2 and Table 2). No significant difference was observed between IVA and IVR in patients without AMI or ischemic stroke history (adjusted HR: 0.87, 95% CI: 0.53–1.42; $P = .583$; Table 2). In patients aged >65 years, the IVA group showed a significantly lower risk than the IVR group did (adjusted HR: 0.64, 95% CI: 0.45–0.92; Fig 3). In the patients without CAD, the IVA group had a significantly lower risk than the IVR group did (adjusted HR: 0.59, 95% CI: 0.37–0.95). The IVA group exhibited a higher but statistically nonsignificant risk of ATEs in patients with CAD (adjusted HR: 1.73, 95% CI: 0.72–4.17). In patients without a history of glaucoma, the glaucoma risk was not significantly different between the IVA group and the IVR group (adjusted HR: 0.63, 95% CI: 0.37–1.06, $P = .084$; Table 3).

## Discussion

### Summary of results

In this study, we presented a population-based incidence of major ATEs and glaucoma in patients who received IVA compared with those who received IVR. The results indicated that the aforementioned side effects did not differ between the two anti-VEGF agents.

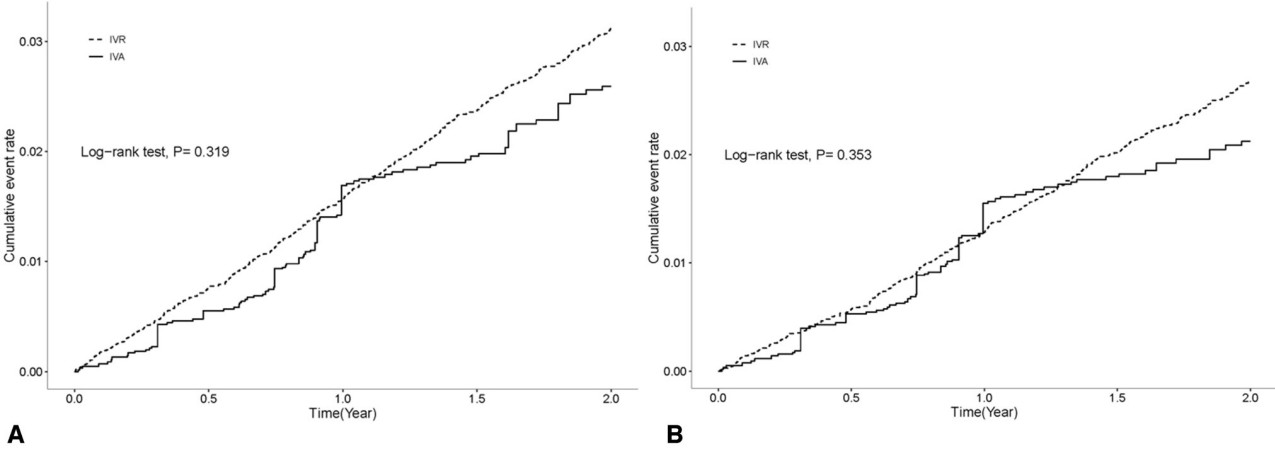

**Fig 2. Kaplan–Meier Failure Curve of ATEs (IS and AMI Hospitalization) in patients receiving IVA injection compared with those receiving IVR injection: (A) Overall cohort (B) Cohort without a history of AMI or IS.** AMI = acute myocardial infarction; ATE = arterial thromboembolic event; IS = ischemic stroke; IVA = intravitreal aflibercept; IVR = intravitreal ranibizumab.

## Risk of ATEs

Our study results are consistent with a retrospective, Asian population cohort study conducted in Singapore, which revealed similar risks of ATEs between intravitreal anti-VEGF agents for various diseases [25]. The Singapore study reported two thromboembolic events in the ranibizumab group (0.06%) and none in the aflibercept group. These two thromboembolic events occurred in patients aged >65 years, which correlated with our subgroup analyses. However, the number of events was small because this study included ATEs that occurred within 1 month after IVI, while our study recorded ATEs for up to 2 years. In terms of ATEs in patients receiving anti-VEGF agents, our study reported slightly higher incidence than that of the Singapore study population but remarkably lower incidence than that of the United States study population, which revealed a cumulative risk of 7.2% for stroke and 6.1% for MI [26, 27]. These two study cohorts were different from our study cohort with respect to the choice of anti-VEGF agents; the patients received intravitreal bevacizumab in most cases. Moreover, the follow-up duration was 2 years in our study and the Singapore study, whereas it was 5 years in

**Table 2. Incidence (per 100 PY) and adjusted HR of ATEs during 1- and 2-year follow-up periods.**

| Cohort | Follow-up period | Treatment | No. of events | PY | Incidence (95% CI) | Adjusted*HR (95% CI) | P |
|---|---|---|---|---|---|---|---|
| **Overall** | **Within 1 year** | IVR | 240 | 15 179 | 1.58 (1.39–1.79) | 1.00 (Ref.) | |
| | | IVA | 53 | 3658 | 1.45 (1.09–1.86) | 0.92 (0.55–1.54) | .739 |
| | **Within 2 years** | IVR | 252 | 28449 | 1.59 (1.45–1.74) | 1.00 (Ref.) | |
| | | IVA | 82 | 5905 | 1.38 (1.10–1.70) | 0.86 (0.57–1.32) | .498 |
| **Cohort Without a History of AMI or IS** | **Within 1 year** | IVR | 185 | 14 332 | 1.29 (1.12–1.49) | 1.00 (Ref.) | |
| | | IVA | 45 | 3473 | 1.30 (0.97–1.73) | 1.00 (0.56–1.81) | .991 |
| | **Within 2 years** | IVR | 367 | 26 940 | 1.36 (1.23–1.51) | 1.00 (Ref.) | |
| | | IVA | 67 | 5610 | 1.19 (0.94–1.52) | 0.87 (0.53–1.42) | .583 |

*Adjusted HR was calculated using Cox proportional hazard analysis adjusted for all variables listed in Table 1.

AMI = acute myocardial infraction; ATE = arterial thromboembolic event; CI = confidence interval; HR = hazard ratio; IS = ischemic stroke; IVA = intravitreal aflibercept; IVR = intravitreal ranibizumab; PY = person-years. Ref. = reference

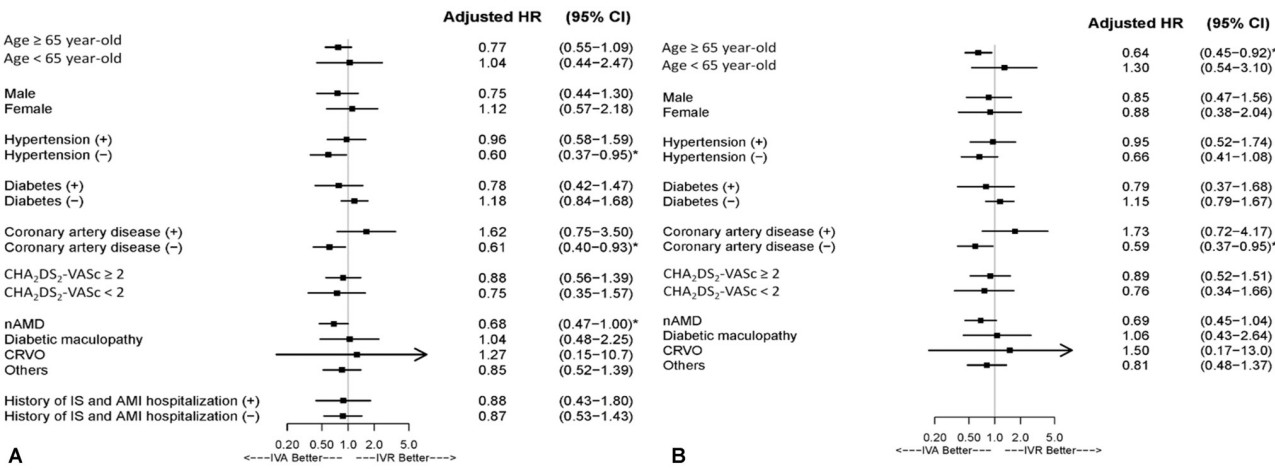

**Fig 3. Subgroup Analysis of Adjusted HR of ATEs (IS or AMI Hospitalization) of patients receiving IVA injection compared with those receiving IVR injection: (A) Overall Cohort (B) Cohort without a history of AMI or IS.** AMI = acute myocardial infarction; ATE = arterial thromboembolic event; HR = hazard ratio; IS = ischemic stroke; IVA = intravitreal aflibercept; IVR = intravitreal ranibizumab.

the US study. Two large, double-masked, randomized controlled trials (VEGF Trap-Eye: Investigation of Efficacy and Safety in Wet AMD [VIEW 1, VIEW 2]) showed similar efficacy and safety outcomes in the IVA and IVR groups [28]. Our study and the VIEW trials both included patients aged >50 years without prior anti-VEGF therapy and with ranibizumab (0.5 mg) or aflibercept (2.0 mg) treatment in the study arms; however, the VIEW trials evaluated only patients with nAMD, whereas we included patients with various retinal diseases, including nAMD, DME, mCNV, and RVO. Male predominance was noted in our cohort, whereas female predominance was noted in the VIEW studies, and the majority of the participants (85%) in the VIEW trials were white. Despite the differences, our study finding is consistent with those of these clinical trials in that systemic ATEs in patients with retinal diseases did not significantly differ between the two anti-VEGF agents.

DRCR.net conducted a comparative effectiveness trial comparing aflibercept, bevacizumab, and ranibizumab in DME treatment [2]. The systemic APTC rates were higher in the ranibizumab group through 2 years, particularly in patients with a history of such events. In the present study, the incidence of ATEs in the DME and RVO groups was much higher than that in the nAMD group (S2 Table). This finding is similar with one previous study, which revealed RVO and DME patients showed higher incidence rates of serious adverse events with anti-VEGF therapy compared with nAMD patients [29]. Serious systemic adverse events have been commonly reported in DME trials, as one would expect in an at-risk diabetic population [30]. The present study showed the risk of ATEs in DME patients did not significantly differ between

**Table 3. Incidence (per 100 PY) and adjusted HR of glaucoma during 1-year and 2-year follow-up periods.**

| Follow-up period | Treatment | No. of events | PY | Incidence (95% CI) | Adjusted*HR (95% CI) | P |
|---|---|---|---|---|---|---|
| **Within 1 year** | IVR | 103 | 14 760 | 0.70 (0.58–0.85) | 1.00 (Ref.) | |
| | IVA | 18 | 3560 | 0.51 (0.32–0.80) | 0.73 (0.39–1.37) | .328 |
| **Within 2 years** | IVR | 170 | 27 416 | 0.62 (0.53–0.72) | 1.00 (Ref.) | |
| | IVA | 23 | 5666 | 0.40 (0.26–0.59) | 0.63 (0.37–1.06) | .084 |

*Adjusted HR was calculated using Cox proportional hazard analysis adjusted for all variables listed in Table 1.

CI = confidence interval; HR = hazard ratio; IVA = intravitreal aflibercept; IVR = intravitreal ranibizumab; PY = person-years.

the IVA and IVR groups, but we did not specifically compare the risks of ATEs in DME patients with a history of APTC events. Further studies are required to evaluate the safety of these two agents for DME patients with a prior history of APTC events.

Our subgroup analyses revealed that patients with CAD showed a higher ATE risk, whereas those without CAD showed a lower ATE risk in the IVA group than in the IVR group. CAD is related to ischemic heart disease, which can lead to AMI. Moreover, CAD is considered a risk factor for cardiotoxicity when a patient receives systematic VEGF inhibitor therapy, such as bevacizumab and sunitinib [31]. Intravitreal anti-VEGF agents could enter the blood stream after intraocular injection, reaching detectable levels in systemic circulation and thereby leading to cardiovascular adverse events [32]. An in vitro study demonstrated that ranibizumab and aflibercept both markedly increased atherosclerosis-associated inflammatory mediators on coronary artery endothelial cells, with aflibercept being significantly more proinflammatory than ranibizumab [33]. Moreover, a study showed that systemic exposure is significantly greater with aflibercept than ranibizumab after intraocular injection [34]. Thus, patients with CAD tended to have more ATEs with IVA than with IVR. Nevertheless, no in vivo study has compared the ATEs between these two treatments specifically in patients with CAD. Aside from CAD, age was an important factor. The elderly already heightened cardiovascular risk, and the average rate of ATEs increased as the age increased [35]. Our large population study demonstrated that patients aged ≥65 years had significantly higher ATE risk in the IVR group than in the IVA group.

## Risk of glaucoma

Our results revealed no statistically significant differences in the risk of glaucoma between the IVA and IVR groups. An acute elevation in IOP after intravitreal injection has been reported in several studies [36–38], and it has been suggested that the repeated transient elevations in IOP after intravitreal injection could be correlated with an increased incidence of glaucoma [39]. Elevations in IOP may become frequent with receiving long term intravitreal anti-VEGF injections, and previous studies have demonstrated that the development or progression of glaucoma is associated with intravitreal injections [40, 41]. The risk of glaucoma after intravitreal injections is concerned because patients with chronic retinal diseases may require multiple injections over time. The injection procedure itself does not significantly vary among anti-VEGF agents, so it was postulated that the agent itself may be a factor related to IOP elevations and glaucoma development [42]. Anti-VEGF agents, or their excipients, may trigger inflammation or immunological reactions, and this may affect aqueous humor production or outflow pathways, such as the trabecular meshwork [42]. Although inflammation more commonly occurs in patients treated with aflibercept, previous studies did not indicate the risk of glaucoma was higher in aflibercept groups than ranibizumab groups [43, 44]. In a post-hoc analysis of data from the IRIS registry, Atchison et al. found that an increase in IOP in 1.9%, 2.8%, and 2.8% of patients treated with aflibercept, ranibizumab, and bevacizumab, respectively [45]. This increase in IOP was higher than in untreated fellow eyes in the bevacizumab and ranibizumab groups but not the aflibercept group. In addition, two studies reported that the retinal never fiber layer thickness did not differ significantly between aflibercept and ranibizumab groups [46, 47]. Our result also showed the risks of glaucoma was not statistically different between the IVA and IVR groups, but we excluded the patients with glaucoma history from these groups. Patients with preexisting glaucoma may be more susceptible to further retinal nerve fiber injury after repeated post-injection IOP spikes [48]. Additional studies are required to establish the link between the long-term risk of glaucomatous optic neuropathy and the use of anti-VEGF agents in patients with preexisting glaucoma.

## Strengths and limitations

The study patients were collected from single-payer health insurance claim data that can reduce the underpowered study bias when evaluating rare adverse events. Moreover, the study cohort included patients who are usually excluded from clinical trials which represents real-world circumstances. Finally, we used strict criteria to select our sample to increase the validity of the findings. This study has some limitations. First, the population data were derived from Taiwan, and hence, the results might not be generalizable to other ethnic groups. In addition, we excluded the patients below 50 years of age because the risk of ATEs is low in patients younger than 50 years old. The results might not be generalizable to the young population. Second, the study was designed retrospectively, so caution should be exercised in drawing inferences. Third, limited subgroup analyses could be performed because the claims database did not include the records of VA, anatomical features, optical coherence tomography, visual field, or intravenous fluorescein angiography results. Moreover, diagnoses of retinal disease might not have been completely identified because the NHIRD only provides the data of the first three diagnoses for each outpatient claim. The claim database did not provide the exclusion criteria for each doctor. Patients with poor systemic profiles might have been excluded from the study group because of the high risks of systemic adverse events of the treatment. Finally, IPTW is increasingly used to estimate the effects of exposures using observational data; however, IPTW could not control unknown confounding factors.

## Conclusion

This retrospective population-based cohort study compared the risk of thromboembolic events between IVA with IVR and demonstrated no significant difference between aflibercept and ranibizumab. Furthermore, glaucoma risk was not significantly different between the studied treatments. However, aflibercept was associated with fewer thromboembolic events in elderly patients; ranibizumab was associated with fewer thromboembolic events in CAD patients. These finding can serve as a reference for decision making regarding the choice of anti-VEGF agents for patients with high cardiovascular risk. Future work is needed to compare these two agents over a long follow-up period.

## Supporting information

**S1 Table. ICD-9/10-CM and ATC code for drug.**
(DOCX)

**S2 Table. Incidence (per 100 PY) and adjusted HR of ATEs among retinal disease subgroups within 2-year follow-up periods.**
(DOCX)

## Acknowledgments

This manuscript was edited by Wallace Academic Editing.

## Author Contributions

**Conceptualization:** Yin-Hsi Chang, Li-Nien Chien, I-Chan Lin.

**Data curation:** Yin-Hsi Chang, Li-Nien Chien, Wan-Ting Chen, I-Chan Lin.

**Formal analysis:** Wan-Ting Chen.

**Investigation:** Wan-Ting Chen.

**Methodology:** Li-Nien Chien, Wan-Ting Chen.

**Software:** Li-Nien Chien.

**Supervision:** Li-Nien Chien, I-Chan Lin.

**Writing – original draft:** Yin-Hsi Chang, I-Chan Lin.

**Writing – review & editing:** I-Chan Lin.

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
