## [Decision Letter · Decision Letter 0]

3 Nov 2020

PONE-D-20-25455

Arterial Thromboembolic Events and Glaucoma Risk with Ranibizumab Versus Aflibercept Intravitreous Injection : A Nationwide Population‐Based Cohort Study

PLOS ONE

Dear Dr. Lin,

Thank you for submitting your manuscript to PLOS ONE. After careful consideration, we feel that it has merit but does not fully meet PLOS ONE’s publication criteria as it currently stands. Therefore, we invite you to submit a revised version of the manuscript that addresses the points raised during the review process.

We look forward to receiving your revised manuscript.

Kind regards,

Vikas Khetan, MD

Academic Editor

PLOS ONE

Additional Editor Comments:

While the study has some merits as pointed by the 2 reviewers, the questions raised by them are pertinent.

Journal Requirements:

4. Please include your tables as part of your main manuscript and remove the individual files. Please note that supplementary tables should be uploaded as separate "supporting information" files.

Reviewers' comments:

Reviewer's Responses to Questions

**Comments to the Author**

1. Is the manuscript technically sound, and do the data support the conclusions?

Reviewer #1: Yes

Reviewer #2: Yes

2. Has the statistical analysis been performed appropriately and rigorously? 

Reviewer #1: Yes

Reviewer #2: Yes

3. Have the authors made all data underlying the findings in their manuscript fully available?

Reviewer #1: Yes

Reviewer #2: Yes

4. Is the manuscript presented in an intelligible fashion and written in standard English?

Reviewer #1: Yes

Reviewer #2: Yes

5. Review Comments to the Author

Reviewer #1: Very nicely written manuscript and is an important information.

It would be good to analyse the retinal disease subgroups, Vein occlusion Vs AMD Vs DME, as some of these have associated systemic diseases more prone for thromboembolic risk.

There are inherent flaws due to the retrospective design, however, as it is nationwide data, it outweighs this. More details on the source of data would add value.

Reviewer #2: This is an important study and has been well executed and written. Following are some suggestions which should be taken into account while revising the manuscript.

MAJOR COMMENTS

The data of thromboembolic adverse events of aflibercept and ranibizumab from the protocol T of drcr.net should be mentioned in the Introduction and Discussion section.

MINOR COMMENTS

Title should be changed to following: Comparison of risks of Arterial Thromboembolic Events and Glaucoma with Ranibizumab and Aflibercept Intravitreous Injection: A Nationwide Population‐Based Cohort Study

Line 137: “missing sex information” in the exclusion criteria should be changed to “missing gender data”

What does the hyperlink in Line 152 refer to?

The section of strength and limitations should be shortened.

6. PLOS authors have the option to publish the peer review history of their article (what does this mean?). If published, this will include your full peer review and any attached files.

Reviewer #1: **Yes: **Rajiv Raman, MS, DNB, FRCS'Ed, Hon DSc

Reviewer #2: No

---

## [Author Response · Author response to Decision Letter 0]

18 Dec 2020

Dear Editor: 

We had revised the manuscript as your suggestion.We had revised the format of the manuscript to meet PLOS ONE's style requirement. In addition, we put the table into the main manuscript.We also checked the Orcid id and added some information on ethics statement. Thank you very much.

Response to reviewers:

Reviewer #1: 

1. Very nicely written manuscript and is an important information.

Response: Thank you very much. 

2. It would be good to analyse the retinal disease subgroups, Vein occlusion Vs AMD Vs DME, as some of these have associated systemic diseases more prone for thromboembolic risk.

Response: Thank you for your suggestion. As you mentioned, some of these disease has been shown associated with systemic diseases, especially DME and CRVO. According to our analysis (S2 Table), we found that the risk of ATE after anti-VEGF therapy was higher in the DME group and RVO group. In the revised manuscript, we provided the analysis to show the risk of ATE in the different retinal disease groups (S2 Table) and also added the discussion of the risk for ATE in different disease groups as your suggestion. (Page 18, Discussion Section, line 294- 299)

S2 Table. Incidence (Per 100 PY) and adjusted HR of ATE among different retinal disease groups within two-year follow-up periods 

Retinal disease groups Treatment No. of

ATE PY Incidence

(95% CI) Adjusted*HR 

(95% CI) P

nAMD IVR 169 13,584 1.24 (1.07-1.45) 1.00 (Ref.) 

 IVA 24 2,808 8.5 (0.55-1.23) 0.68 (0.47-1.00) 0.048

DME IVR 207 8,034 2.58 (2.24-2.94) 1.00 (Ref.) 

 IVA 42 1,495 2.82 (2.08-3.80) 1.04 (0.48-2.25) 0.916

RVO IVR 5 212 2.17 (0.77-4.84) 1.00 (Ref.) 

 IVA 2 63 3.16 (0.39-8.88) 1.27 (0.15-10.7) 0.828

Others IVR 71 6,619 1.08 (0.85-1.35) 1.00 (Ref.) 

 IVA 14 1,538 0.90 (0.50-1.45) 0.85 (0.52-1.39) 0.520

*Adjusted HR was performed by Cox proportional hazard adjusted for all variables listed in Table 1. 

Abbreviation: ATE = acute thromboembolic event; DME= Diabetic maculopathy; HR = hazard ratio; IVR = intravitreal ranibizumab; IVA = intravitreal aflibercept; PY=person year; Ref.=reference

3. There are inherent flaws due to the retrospective design, however, as it is nationwide data, it outweighs this. More details on the source of data would add value.

Response: Thank you. We provided more details on the source of data in the revised manuscript as your suggestion (Page 7, Materials and methods section, line 127-142):

Baseline characteristics of patients were derived from the National Health Insurance Research Data (NHIRD), a reimbursement claims database that covers 99% of the residents in Taiwan enrolled in the National Health Insurance (NHI) program. NHI is a single-payer health insurance system and has a contract with most healthcare providers in Taiwan. NHI provides a universal coverage that covers all necessary medical expenses including outpatient visits, the inpatient system, prescriptions, treatment with traditional Chinese medicine, dental services, operations, and investigations such as X-rays or magnetic resonance imaging. It is mandatory for care providers to upload the claims data from each service to the National Health Insurance Administration. Therefore, the NHIRD files include inpatient, outpatient, and pharmaceutical reimbursement claims, and the disease diagnosis was coded using the International Classification of Diseases, 9th Revision, Clinical Modification (ICD-9-CM) and 10th revision (ICD-10-CM) codes after the year 2016. Because patient data are deidentified in the NHIRD before release to researchers, the requirement for informed consent under the full review process of the TMU-JIRB was waived.

Reviewer #2: This is an important study and has been well executed and written. Following are some suggestions which should be taken into account while revising the manuscript.

MAJOR COMMENTS

4. The data of thromboembolic adverse events of aflibercept and ranibizumab from the protocol T of drcr.net should be mentioned in the Introduction and Discussion section.

Response:

Thank you for your suggestion. We had added the data from the protocol T of DRCR.net in the Introduction (Page 6, line 101-112) and Discussion (Page 18, line 292 -302). 

MINOR COMMENTS

5. Title should be changed to following: Comparison of risks of Arterial Thromboembolic Events and Glaucoma with Ranibizumab and Aflibercept Intravitreous Injection: A Nationwide Population‐Based Cohort Study

Response: Thanks for your suggestion. We changed the title as your suggestion. Please check the title in the revised manuscript. 

6. Line 137: “missing sex information” in the exclusion criteria should be changed to “missing gender data”

Response: Thanks for your suggestion. We changed the missing sex information into missing gender information. Please check the revised manuscript (Fig 1, line 5). 

7. What does the hyperlink in Line 152 refer to?

Response: We provided the references for the description of the CHA2DS2score. (Page 26 , line 431-433, Reference 17 )

8. The section of strength and limitations should be shortened.

Response: Thanks for your suggestion. We have shorten the strength and limitation section in the revised manuscript your suggestion.

(Page 21 , line 334)

4.4 Strengths and Limitations. The study patients was collected from a single-payer health insurance claim data that can reduce the underpowered study bias when evaluating rare adverse events. Moreover, the study cohort included patients who are usually excluded from clinical trials which represents real-world circumstances. Finally, we used strict criteria to select our sample that increase validity of the findings. This study has some limitations. First, the population data were derived from Taiwan, and hence, the results might not be generalizable to other ethnic groups. Second, the study was designed retrospectively, so caution should be exercised in drawing inferences. Third, limited subgroup analyses could be performed because the claims database did not include the records of visual acuity, anatomical features, optical coherence tomography, or intravenous fluorescein angiography results. Also, the diagnoses of retinal disease might not have been completely identified because the NHIRD only provided the data of the first 3 diagnoses for each outpatient claim. Finally, IPTW is increasingly used to estimate the effects of exposures using observational data; however, IPTW could not control unknown confounding factors.

---

## [Decision Letter · Decision Letter 1]

23 Feb 2021

PONE-D-20-25455R1

Comparison of risks of Arterial Thromboembolic Events and Glaucoma with Ranibizumab and Aflibercept Intravitreous Injection: A Nationwide Population‐Based Cohort Study

PLOS ONE

Dear Dr. Lin,

Thank you for submitting your manuscript to PLOS ONE. After careful consideration, we feel that it has merit but does not fully meet PLOS ONE’s publication criteria as it currently stands. Therefore, we invite you to submit a revised version of the manuscript that addresses the points raised during the review process.

We look forward to receiving your revised manuscript.

Kind regards,

Vikas Khetan, MD

Academic Editor

PLOS ONE

Journal Requirements:

Additional Editor Comments (if provided):

The paper is relatively well written

Reviewers' comments:

Reviewer's Responses to Questions

**Comments to the Author**

1. If the authors have adequately addressed your comments raised in a previous round of review and you feel that this manuscript is now acceptable for publication, you may indicate that here to bypass the “Comments to the Author” section, enter your conflict of interest statement in the “Confidential to Editor” section, and submit your "Accept" recommendation.

Reviewer #1: All comments have been addressed

Reviewer #2: All comments have been addressed

2. Is the manuscript technically sound, and do the data support the conclusions?

Reviewer #1: Yes

Reviewer #2: Yes

3. Has the statistical analysis been performed appropriately and rigorously? 

Reviewer #1: Yes

Reviewer #2: Yes

4. Have the authors made all data underlying the findings in their manuscript fully available?

Reviewer #1: Yes

Reviewer #2: Yes

5. Is the manuscript presented in an intelligible fashion and written in standard English?

Reviewer #1: Yes

Reviewer #2: Yes

6. Review Comments to the Author

Reviewer #1: (No Response)

Reviewer #2: Authors have modified the manuscript as suggested. There is no comments or changes to be made at this stage.

7. PLOS authors have the option to publish the peer review history of their article (what does this mean?). If published, this will include your full peer review and any attached files.

Reviewer #1: **Yes: **RAJIV RAMAN

Reviewer #2: No

---

## [Author Response · Author response to Decision Letter 1]

6 Apr 2021

Thank you for the editors and the reviewers’ comments. We revised the manuscript as your suggestion:

1. Please review your reference list to ensure that it is complete and correct. 

Response: The references of manuscript had been revised as follows:

Reference 6 changed to Chou Y-B, Chen M-J, Lin T-C, Chen S-J, Hwang D-K. Priority options of anti-vascular endothelial growth factor agents in wet age-related macular degeneration under the National Health Insurance Program. Journal of the Chinese Medical Association. 2019;82(8), which is relevant to the content.(Page 6, line 87;Page 25, line375, Reference 6)

Some errors were noted on the Reference 11,12,13,14,15,16 and 17. We had revised them. Thank you for your valuable suggestion.

(Page 7, line 105, line 111; Page 8, line 131;Page 10, line 159, line 169)

Reference 30 changed to Ahn J, Jang K, Sohn J, Park JI, Hwang DD-J. Effect of intravitreal ranibizumab and aflibercept injections on retinal nerve fiber layer thickness. Scientific Reports. 2021;11(1):5010. doi: 10.1038/s41598-021-84648-1 (Page 21, line 329;Page 29, line 454, Reference 30)

2. If you need to cite a retracted article, indicate the article’s retracted status in the References list and also include a citation and full reference for the retraction notice.

Response: All references have been checked with the retraction database, and no retracted reference article was identified.

3. We corrected the name of first name to Yin-Hsi Chang.(Page 1,line 5)

4. We added the description of the abbreviation “FDA” (Page 5, line 77)

---

## [Decision Letter · Decision Letter 2]

29 Jun 2021

PONE-D-20-25455R2

Comparison of risks of Arterial Thromboembolic Events and Glaucoma with Ranibizumab and Aflibercept Intravitreous Injection: A Nationwide Population‐Based Cohort Study

PLOS ONE

Dear Dr. Lin,

Thank you for submitting your manuscript to PLOS ONE. After careful consideration, we feel that it has merit but does not fully meet PLOS ONE’s publication criteria as it currently stands. Therefore, we invite you to submit a revised version of the manuscript that addresses the points raised during the review process.

We look forward to receiving your revised manuscript.

Kind regards,

Vikas Khetan, MD

Academic Editor

PLOS ONE

Journal Requirements:

Reviewers' comments:

Reviewer's Responses to Questions

**Comments to the Author**

1. If the authors have adequately addressed your comments raised in a previous round of review and you feel that this manuscript is now acceptable for publication, you may indicate that here to bypass the “Comments to the Author” section, enter your conflict of interest statement in the “Confidential to Editor” section, and submit your "Accept" recommendation.

Reviewer #2: All comments have been addressed

Reviewer #3: (No Response)

2. Is the manuscript technically sound, and do the data support the conclusions?

Reviewer #2: Yes

Reviewer #3: Partly

3. Has the statistical analysis been performed appropriately and rigorously? 

Reviewer #2: Yes

Reviewer #3: Yes

4. Have the authors made all data underlying the findings in their manuscript fully available?

Reviewer #2: Yes

Reviewer #3: Yes

5. Is the manuscript presented in an intelligible fashion and written in standard English?

Reviewer #2: Yes

Reviewer #3: No

6. Review Comments to the Author

Reviewer #2: Authors have revised the manuscript as suggested during previous revision. No additional comments at this stage of revision.

Reviewer #3: Dear author, I am glad to have reviewed this study. As the intravitreal injections are being administered in high numbers all across the world, we need to have a real world data about the possible systemic and ocular side effects. Although you have tried to bring out the safety profile in a subset of patients, I have concerns which need to be addressed adequately.

1. Page 11, line 78: FDA approval is given for a particular disease, for eg, Ranibizumab was approved for treating wet AMD in 2006, whereas Aflibercept was approved in 2011 for the same disease. Are you pointing out that Taiwan FDA approved these two drugs for all the diseases mentioned in the study together? Please modify your statements here.

2. There are gross Grammatical errors, please check editing as well.

3. Please explain: ICDR-9 and 10 CM coding was acceptable after 2016, whereas you selected the patients between 2011 and 2016. Does that mean that the ICDR coding was modified after 2016 for the patients seen before that year? Or there is a discrepancy in diagnosis and selection bias here because of the different ICDR coding definitions?

4. Please include the following in exclusion/inclusion criteria: patients with poor systemic profile, and what was the minimum interval between the last episode of cardiovascular event and injection AntiVEGF?

5. Please explain this: You have excluded patients below 50 years of age. Yet, for the Inverse Probability Treatment Weighting, you have used CHA2DS2 scoring system which scores patients above 65 and above 75 years. This puts the patients between 50-65 years as confounded due to age, as they could not be scored for weight adjustments.

6. To exclude the patients who are at increased risk of Glaucoma, the exclusion criteria must exclude pre-existing glaucoma/axial length variations/ phakic status/ patients status post laser/ vitrectomy and steroid use. These are important to differentiate between patients who developed glaucoma due to a particular AntiVEGF agent versus those who had a high tendency to develop glaucoma after any injection.

7. You have included the incidence of adverse events occurring even after 2 years following the last AntiVEGF agent use. This would be inappropriate as the adverse event will definitely not reflect a direct cause and effect relationship with the AntiVEGF agent.

8. You have done an excellent job in S2 as the patients with DME and RVO should be analyzed separately from those with wet AMD, as the systemic profile of such patients may already be compromised to a great extent prior to AntiVEGF use.

9. In subgroup analysis in patients who developed glaucoma, please include the following: when was IOP measured after the injection, what was the IOP spike, how many AGMs were used to control the IOP and was the IOP controlled or not, when compared between the ranibizumab and aflibercept groups?

Thank you

7. PLOS authors have the option to publish the peer review history of their article (what does this mean?). If published, this will include your full peer review and any attached files.

Reviewer #2: No

Reviewer #3: No

---

## [Author Response · Author response to Decision Letter 2]

6 Aug 2021

For Journal requirement: 

Reply:

1. We had sent the manuscript to a professional English editing service to revise all the typographical or grammatical errors. 

2. We have verified all the references on the website Retraction Watch.

3. Reference 30 changed to Ahn J, Jang K, Sohn J, Park JI, Hwang DD-J. Effect of intravitreal ranibizumab and aflibercept injections on retinal nerve fiber layer thickness. Scientific Reports. 2021;11(1):5010. doi: 10.1038/s41598-021-84648-1

 (page 22, line 331-332;page 29, line 463, reference 30)

For reviewer #2: Thank you for your comments.

For reviewer #3 

1. Page 11, line 78: FDA approval is given for a particular disease, for eg, Ranibizumab was approved for treating wet AMD in 2006, whereas Aflibercept was approved in 2011 for the same disease. Are you pointing out that Taiwan FDA approved these two drugs for all the diseases mentioned in the study together? Please modify your statements here.

Reply: Thank you for your comments.

1. Similar to the United States, the use of anti-VEGF agents was first approved for nAMD treatment in Taiwan. Ranibizumab and aflibercept were approved for treating nAMD by the Taiwan FDA in 2009 and 2011, respectively. Their use for other indications including DME was approved in subsequent years. Currently, these agents can be used in the treatment of nAMD, DME, CRVO, BRVO, PCV and myopic CNV. We have made revisions as follow:

Ranibizumab as a treatment for retinal neovascularization diseases such as neovascular age-related macular degeneration (nAMD) was approved for use by the US Food and Drug administration (FDA) in June 2006 and by the Taiwan FDA since 2009. Aflibercept (Eylea, Regeneron, Tarrytown, PA, USA and Bayer HealthCare, Berlin, Germany) was also approved by the US FDA in November 2011 and by the Taiwan FDA in June 2013 for treating nAMD, and its use has rapidly increased since then. (page 5, line 75-81)

2. There are gross Grammatical errors, please check editing as well.

Reply: We have availed a professional English editing service to recheck the manuscript and revise all the grammatical errors of the manuscript.

page 3: line 41, 43: add “the”, line 50 add “those”

page 4: line 52: add “the”

 line 54: “was not significantly different” change to “did not significantly differ”

 line 57:” our study suggested that that these two intravitreal agents had no significant difference ” change to “No significant differences in the risk of major ATEs and glaucoma were found between ranibizumab and aflibercept,”

page 5: line 83 add “that of” 

page 6: line 87: add “related to the use of these agents”

 line 95: add “who received”

 line98: add “in the treatment of”

line100: “ATPC” change to “APTC”

page 7: line 109: “quickly” change to “rapidly”

line 109: add “and”

page 8: line125: add “The” “program”

line126: add “that”

line128: add “s”

page 9: line 135: add “The”

line 145: “consisted of“ change to “comprised”

page 13: line 206: add “occurred”

 line 209: add “The”

 line 211: add “The”

line 215: add “the”

line 217 and line 218: add “USA”

page 14: line 232,234,235: add “the”

page 18; line 264: add “s”

 line 266: “these two anti-VEGF agents exhibited no difference in terms of the aforementioned side effects.” change to “the aforementioned side effects did not differ between the two anti-VEGF agents”

page 19; line 280: add “cohort”

line: 284: add “the”

page 20: line 296. add “the”

 line 298: add “s”

line 301. add “the”

line 306: add “the”

page 21:line320:add “the”

 line 322, 324: add “the”

 line 327: add “an”

Thank you for your suggestion.

3. Please explain: ICDR-9 and 10 CM coding was acceptable after 2016, whereas you selected the patients between 2011 and 2016. Does that mean that the ICDR coding was modified after 2016 for the patients seen before that year? Or there is a discrepancy in diagnosis and selection bias here because of the different ICDR coding definitions?

Reply: Thank you for your suggestion,

1. Because the prescriptions of ranibizumab and aflibercept were first reimbursed by the National Health Insurance in Taiwan on January 1, 2011, and August 1, 2014, respectively, we selected a study period of 2011–2016. We selected 2017 as the follow-up year to ensure that at least 1 year of follow-up data were available for all eligible patients (page 9, line 148–150). 

2. The ICDR coding was modified after 2016 for the patients seen before that year, and there might be a discrepancy in diagnosis because of the different ICDR coding. However, we separated the study patients into two groups based on the prescription claims, not based on the diagnostic codes. Because we used the same methods and medical coding systems to compare the outcomes between the IVR and IVA groups, no selection bias was noted between the two groups.

3. We have presented all the ICDR codes in Table S1. Care providers must upload the medical records of patients’ reimbursement applications for ranibizumab and aflibercept. Approval for ranibizumab and aflibercept reimbursement is not based on medical coding alone but also includes a review of the patient’s medical records by a retinal specialist. The review process increases the validity of the diagnosis. We have added the relevant explanation in the revised manuscript.(page 8, line 131–134)

 4. Please include the following in exclusion/inclusion criteria: patients with poor systemic profile, and what was the minimum interval between the last episode of cardiovascular event and injection AntiVEGF?

Reply: Thank you for your valuable comments.

1. In our study, we used inverse probability treatment weighting based on the propensity score to balance the baseline characteristics between the two groups. Although we did not specifically include or exclude patients with poor systemic profiles, we did separately analyze the risk in patients with and without a history of AMI or stroke. (page 17. Table 2)

2. Patients with poor systemic profiles might be excluded from the treatment because of the high risks of systemic adverse effects. However, this claim-based data study was based on the disease diagnostic codes and prescription claims. The detailed information of exclusion criteria for each doctor is not provided in this claims database. We have added the relevant explanation in the revised manuscript (page 23, line 350-352) However, we believe that this limitation might not bias our results because we used the same methods to compare the outcomes between the two groups.

3. We separately analyzed the data in patients with and without a history of AMI or stroke, and we did not specifically exclude patients with recent cardiovascular event. The definition of cardiovascular events was based on the diagnostic codes and hospitalization (page12, line 187-190). A history of AMI or stroke was defined as the occurrence of AMI or stroke events before the index date (anti-VEGF injection). We did not specifically record the last episode of cardiovascular events of each patient, and a further study for patients with recurrent stroke or AMI events after intravitreal injection will be needed in the future.

4 Please explain this: You have excluded patients below 50 years of age. Yet, for the Inverse Probability Treatment Weighting, you have used CHA2DS2 scoring system which scores patients above 65 and above 75 years. This puts the patients between 50-65 years as confounded due to age, as they could not be scored for weight adjustments.

Reply: Thank you for your valuable comments.

1. We excluded patients below 50 years of age because the risk of ATE is low among patients younger than 50 years old. Because both the IVR and IVA groups had the same exclusion criteria, the exclusion of patients younger than 50 years should not have biased our results, although generalizability to young patients was limited. We have added the relevant explanation in the revised manuscript (page 22, line 342-344)

2. We used the CHA2DS2 scoring system as a covariate to adjust for its confounding effect on cardiovascular risk. According to the scoring system, a patient aged between 50 and 64 years is given 0 points. Thus, their scores were included for weight adjustments in the study. We have added this information in the revised manuscript.(page 11,line 170 )

6. To exclude the patients who are at increased risk of Glaucoma, the exclusion criteria must exclude pre-existing glaucoma/axial length variations/ phakic status/ patients status post laser/ vitrectomy and steroid use. These are important to differentiate between patients who developed glaucoma due to a particular AntiVEGF agent versus those who had a high tendency to develop glaucoma after any injection.

Reply: Thanks for your comments.

1. Several studies had reported preexisting glaucoma to be a risk factor for sustained ocular hypertension after AntiVEGF injection1,2.

 In our comparison of the risk of glaucoma between the two groups, we excluded patients with preexisting glaucoma and only compared patients without a history of glaucoma on Table 3. (page 14and 15, line 236–240, and page 17, Table 3) No significant differences in glaucoma development were noted between the two groups. 

2. This claim-based study was based on the claims data of diagnostic codes, prescriptions, and procedures. However, the database did not include the data on axial length, anatomical features, or OCT results. We have described this as a limitation of the study in the revised manuscript.(page 22 and 23 ,line 345-347)

References:

1. Maruyama-Inoue, M., Inoue, T., Mohamed, S. et al. Incidence of elevated intraocular pressure after intravitreal injection in Japanese patients with age-related macular degeneration. Sci Rep 11, 12246 (2021). https://doi.org/10.1038/s41598-021-91832-w

2. Dedania VS, Bakri SJ. SUSTAINED ELEVATION OF INTRAOCULAR PRESSURE AFTER INTRAVITREAL ANTI-VEGF AGENTS: What Is the Evidence? Retina. 2015 May;35(5):841-58. doi: 10.1097/IAE.0000000000000520. PMID: 25905784.

7. You have included the incidence of adverse events occurring even after 2 years following the last AntiVEGF agent use. This would be inappropriate as the adverse event will definitely not reflect a direct cause and effect relationship with the AntiVEGF agent.

Reply: Thank you for your comments,

1. Our patients were included on the date of their first anti-VEGF injection and followed up for up to 2 years. Although a direct causal effect could not be concluded from the present study, we provided long-term real-world data in terms of the cumulative risk.

2. FOCUS, MARINA, ANCHOR, PIER, and SAILOR were trials that assessed the outcomes of ranibizumab treatment at either 1 or 2 years.1-5 Bressler et al. pooled these studies to analyze the CVA risk at 1 year and 2 years.6 Kitchens et al. reviewed 10 phase II and III trials and calculated the risk of ocular and systemic adverse events as events per person-years (PYR) and rate ratios (RR) of aflibercept treatment.7 The study period of that also exceeded 1 year. Our study had a similar follow-up period.

3. An acute spike in IOP elevation after intravitreal injection has been reported in several studies, and even sustained IOP elevation after anti-VEGF injection has been revealed.8-12 However, a short term IOP spike after antiVEGF injection seems not associated with the damage to optic nerve12, and whether ocular hypertension develops into glaucoma requires repeat evaluations over the long term. For instance, Casa et al analyzed the retinal nerve fiber layer thickness changes in nAMD patients receiving anti-VEGF injection over 1 year.13 Filek et al found increasing cup-disc ratio over time in DME patients with anti-VEGF injection in a 2-year analysis.14 Our study was designed to compare the risk of glaucoma between the two groups, and the cases of glaucoma was defined as at least three times of diagnostic codes with antiglaucoma agents (ATC code S01E) claims.(page 12, line 190-191) Therefore, patients with short-term ocular hypertension will be excluded, and a follow-up period of 1-2 years is required to assess whether a significant risk of glaucoma exists.

1. Rosenfeld PJ, Brown DM, Heier JS, Boyer DS, Kaiser PK, Chung CY, Kim RY. Ranibizumab for Neovascular Age-Related Macular Degeneration. New England Journal of Medicine. 2006;355(14):1419-1431.

2. Regillo CD, Brown DM, Abraham P, Yue H, Ianchulev T, Schneider S, Shams N. Randomized, Double-Masked, Sham-Controlled Trial of Ranibizumab for Neovascular Age-related Macular Degeneration: PIER Study Year 1. American journal of ophthalmology. 2008;145(2):239-248.e235.

3. Antoszyk AN, Tuomi L, Chung CY, Singh A. Ranibizumab combined with verteporfin photodynamic therapy in neovascular age-related macular degeneration (FOCUS): year 2 results. American journal of ophthalmology. 2008;145(5):862-874.

4. Brown DM, Kaiser PK, Michels M, Soubrane G, Heier JS, Kim RY, Sy JP, Schneider S. Ranibizumab versus verteporfin for neovascular age-related macular degeneration. The New England journal of medicine. 2006;355(14):1432-1444.

5. Brown DM, Michels M, Kaiser PK, Heier JS, Sy JP, Ianchulev T. Ranibizumab versus verteporfin photodynamic therapy for neovascular age-related macular degeneration: Two-year results of the ANCHOR study. Ophthalmology. 2009;116(1):57-65.e55.

6. Bressler NM, Boyer DS, Williams DF, Butler S, Francom SF, Brown B, Di Nucci F, Cramm T, Tuomi LL, Ianchulev T, Rubio RG. Cerebrovascular accidents in patients treated for choroidal neovascularization with ranibizumab in randomized controlled trials. Retina (Philadelphia, Pa). 2012;32(9):1821-1828.

7. Kitchens JW, Do DV, Boyer DS, Thompson D, Gibson A, Saroj N, Vitti R, Berliner AJ, Kaiser PK. Comprehensive Review of Ocular and Systemic Safety Events with Intravitreal Aflibercept Injection in Randomized Controlled Trials. Ophthalmology. 2016;123(7):1511-1520.

8. Bressler SB, Almukhtar T, Bhorade A, Bressler NM, Glassman AR, Huang SS, Jampol LM, Kim JE, Melia M. Repeated intravitreous ranibizumab injections for diabetic macular edema and the risk of sustained elevation of intraocular pressure or the need for ocular hypotensive treatment. JAMA ophthalmology. 2015;133(5):589-597.

9. Kim JE, Mantravadi AV, Hur EY, Covert DJ. Short-term Intraocular Pressure Changes Immediately After Intravitreal Injections of Anti–Vascular Endothelial Growth Factor Agents. American journal of ophthalmology. 2008;146(6):930-934.e931.

10. Good TJ, Kimura AE, Mandava N, Kahook MY. Sustained elevation of intraocular pressure after intravitreal injections of anti-VEGF agents. The British journal of ophthalmology. 2011;95(8):1111-1114.

11. Tseng JJ, Vance SK, Della Torre KE, Mendonca LS, Cooney MJ, Klancnik JM, Sorenson JA, Freund KB. Sustained increased intraocular pressure related to intravitreal antivascular endothelial growth factor therapy for neovascular age-related macular degeneration. Journal of glaucoma. 2012;21(4):241-247.

12. Bracha P, Moore NA, Ciulla TA, WuDunn D, Cantor LB. The acute and chronic effects of intravitreal anti-vascular endothelial growth factor injections on intraocular pressure: A review. Survey of ophthalmology. 2018;63(3):281-295.

13. Martinez-de-la-Casa JM, Ruiz-Calvo A, Saenz-Frances F, Reche-Frutos J, Calvo-Gonzalez C, Donate-Lopez J, Garcia-Feijoo J. Retinal nerve fiber layer thickness changes in patients with age-related macular degeneration treated with intravitreal ranibizumab. Invest Ophthalmol Vis Sci. 2012;53(10):6214-6218.

14. Filek R, Hooper P, Sheidow TG, Gonder J, Chakrabarti S, Hutnik CM. Two-year analysis of changes in the optic nerve and retina following anti-VEGF treatments in diabetic macular edema patients. Clinical ophthalmology (Auckland, NZ). 2019;13:1087-1096.

8 You have done an excellent job in S2 as the patients with DME and RVO should be analyzed separately from those with wet AMD, as the systemic profile of such patients may already be compromised to a great extent prior to AntiVEGF use.

Reply: Thank you for your comments.

9. In subgroup analysis in patients who developed glaucoma, please include the following: when was IOP measured after the injection, what was the IOP spike, how many AGMs were used to control the IOP and was the IOP controlled or not, when compared between the ranibizumab and aflibercept groups?

Reply: Thank you for your comments.

1. In the manuscript, we have discussed this as a limitation of the claims-based study. The study was based on the claims data of diagnostic codes, prescriptions, and procedures. The database did not include the details of IOP, OCT, or visual field results.(page 22, line 345-347) Therefore, this database did not provide the information whether the IOP was controlled or not. Future work is needed to compare the risks of acute IOP elevation between these two agents, and assess the glaucoma control of these patients.

---

## [Decision Letter · Decision Letter 3]

17 Nov 2021

PONE-D-20-25455R3Comparison of risks of Arterial Thromboembolic Events and Glaucoma with Ranibizumab and Aflibercept Intravitreous Injection: A Nationwide Population‐Based Cohort StudyPLOS ONE

Dear Dr. Lin,

Thank you for submitting your manuscript to PLOS ONE. After careful consideration, we feel that it has merit but does not fully meet PLOS ONE’s publication criteria as it currently stands. Therefore, we invite you to submit a revised version of the manuscript that addresses the points raised during the review process.

We look forward to receiving your revised manuscript.

Kind regards,

Vikas Khetan, MD

Academic Editor

PLOS ONE

Reviewers' comments:

Reviewer's Responses to Questions

**Comments to the Author**

1. If the authors have adequately addressed your comments raised in a previous round of review and you feel that this manuscript is now acceptable for publication, you may indicate that here to bypass the “Comments to the Author” section, enter your conflict of interest statement in the “Confidential to Editor” section, and submit your "Accept" recommendation.

Reviewer #3: All comments have been addressed

Reviewer #4: (No Response)

2. Is the manuscript technically sound, and do the data support the conclusions?

Reviewer #3: Yes

Reviewer #4: Partly

3. Has the statistical analysis been performed appropriately and rigorously? 

Reviewer #3: Yes

Reviewer #4: Yes

4. Have the authors made all data underlying the findings in their manuscript fully available?

Reviewer #3: Yes

Reviewer #4: No

5. Is the manuscript presented in an intelligible fashion and written in standard English?

Reviewer #3: Yes

Reviewer #4: Yes

6. Review Comments to the Author

Reviewer #3: Dear author,

I really appreciate your efforts in modifying the manuscript to publication standards.

Thank you

Reviewer #4: Major comments

In the Introduction authors write:

“However, few studies have compared the risk of elevated IOP between ranibizumab and aflibercept”

Authors need to quote the studies which they mention in this sentence.

Why do authors think that comparing the intraocular pressure rise between ranibizumab and aflibercept was justified? Why do they think that the rise would be different between the two drugs?

In Results section authors write:

“In patients without a history of glaucoma, the 237 glaucoma risk was not significantly different between the IVA group and the IVR group (adjusted HR: 0.63, 95% CI: 0.37–1.06, P= .084; Table 3).

The criteria to define glaucoma have not been mentioned in the Methods section.

This is the single sentence in the Results which deals with intraocular pressure rise aspect in this study. Intraocular pressure monitoring is standard care before and after intravitreal injection. Why do authors chose to report and combine intraocular pressure data alongside systemic thromboembolic adverse events as both are completely unconnected.

Minor Comments

In the Methods authors write:

“Because patient data are deidentified in the NHIRD before release to researchers, the requirement for informed consent under the full review process of the TMU-JIRB was waived.”

Does the data storage policy include patients’ consent permitting use of their deidentified data for research purpose?

“Additionally, the study used the National Death Registry under the regulation of Health and Welfare Data Science Center, Ministry of Health and Welfare, Executive Yuan, Taiwan.”

The use of this data in the study needs to be elaborated.

In Discussion section authors write:

“Besides, the Singapore study was performed in a single center; therefore, the data did not represent the national population”

Do authors really think that this criticism was valid in view of small size and population of city state Singapore?

“The present study showed the risk of ATEs in DME patients did not significantly differ between the IVA and IVR groups, but we did not specifically compare the risks of ATEs in DME patients with a history of such events”

The second part of this sentence is unclear.

The section of Discussion dealing with “Risk of Glaucoma” appears nondescript and not clearly written. It does not drive home the intended message.

7. PLOS authors have the option to publish the peer review history of their article (what does this mean?). If published, this will include your full peer review and any attached files.

Reviewer #3: No

Reviewer #4: **Yes: **KUMAR SAURABH

---

## [Author Response · Author response to Decision Letter 3]

31 Dec 2021

Major comments from the reviewers

1. In the introduction: “However, few studies have compared the risk of elevated IOP between ranibizumab and aflibercept’. Authors need to quote the studies which they mention in this sentence. 

Reply: Thanks for your comments. We have added the references to the following sentence in the revised manuscript : “However, few studies have compared the risk of elevated IOP between ranibizumab and aflibercept [15-17] ”(page 7, line 115-116 ). We also revised the list of references(page 27-28 , line 426-436)

2. Why do authors think that comparing the intraocular pressure rise between ranibizumab and aflibercept was justified? Why do they think that the rise would be different between two drugs?

Reply: Thanks the valuable comment. 

In this study ,we used three claims of diagnosis codes with anti-glaucoma medication to increase the validity of diagnosis. (page 11 , line 201-204).One of the limitations of this study is that we did not get the detail records of IOP, and we only used the diagnosis codes with glaucoma medication prescription to compare the risks of glaucoma(page 23, line 356-357).

Several studies have reported IOP elevation after intravitreal injection,1-3 and the repeated elevations in IOP after intravitreal injections could be correlated with an increased incidence of glaucoma.4,5,6 The risks of glaucoma is concerned because patients with chronic retinal diseases may require multiple intravitreal injections over time. The injection procedure itself does not vary considerably between different anti-VEGF agents7; therefore, we postulated that the agent itself rather than the injection procedure is related to the observed elevations in IOP. Limited numbers of studies have compared the risks of glaucoma between different anti-VEGF agents.7 Therefore, we compared the risks of glaucoma between these two agents.We have added more discussions to the revised manuscript(page 21-22 , line 321-345).

References:

1.Good TJ, Kimura AE, Mandava N, Kahook MY. Sustained elevation of intraocular pressure after intravitreal injections of anti-VEGF agents. The British journal of ophthalmology. 2011;95(8):1111-1114.

2.Tseng JJ, Vance SK, Della Torre KE, Mendonca LS, Cooney MJ, Klancnik JM, Sorenson JA, Freund KB. Sustained increased intraocular pressure related to intravitreal antivascular endothelial growth factor therapy for neovascular age-related macular degeneration. Journal of glaucoma. 2012;21(4):241-247.

3. Bracha P, Moore NA, Ciulla TA, WuDunn D, Cantor LB. The acute and chronic effects of intravitreal anti-vascular endothelial growth factor injections on intraocular pressure: A review. Survey of ophthalmology. 2018;63(3):281-295.

 4. de Vries VA, Bassil FL, Ramdas WD. The effects of intravitreal injections on intraocular pressure and retinal nerve fiber layer: a systematic review and meta-analysis. Sci Rep. 2020;10(1):13248. Published 2020 Aug 6. doi:10.1038/s41598-020-70269-7

5. Du, J. , Patrie, J. T. , Prum, B. E. , Netland, P. A. & Shildkrot, Y. (. (2019). Effects of Intravitreal Anti-VEGF Therapy on Glaucoma-like Progression in Susceptible Eyes. Journal of Glaucoma, 28 (12), 1035-1040. doi: 10.1097/IJG.0000000000001382.

6. Mansoori T, Agraharam SG, Manwani S, Balakrishna N. Intraocular Pressure Changes after Intravitreal Bevacizumab or Ranibizumab Injection: A Retrospective Study. J Curr Ophthalmol. 2021;33(1):6-11. Published 2021 Mar 26. doi:10.4103/JOCO.JOCO_5_20

7. Ramsey, D.J., McCullum, J.C., Steinberger, E.E. et al. Intraocular pressure decreases in eyes with glaucoma-related diagnoses after conversion to aflibercept for treatment-resistant age-related macular degeneration. Eye (2021). https://doi.org/10.1038/s41433-021-01729-1

3. In the result: “In patients without a history of glaucoma, the glaucoma risk was not significantly different between the IVA group and the IVR group (adjusted HR: 0.63, 95% CI: 0.37–1.06, P 239 = .084; Table 3). The criteria to define glaucoma have not been mentioned in the Method section. 

Reply: Thanks for your comments.

In the manuscript, we described the the definition of glaucoma in the Method-Main Outcome Measurements sections (page 11, line 200-202): “Glaucoma was defined as at least three times of diagnoses based on ICD-9-CM or ICD-10-CM diagnostic codes with antiglaucoma agents (ATC code S01E) administration”.The details related to the dignositc codes are provided in the Supporting Information section (S1 Table). In addition, we have added additional relevant information in the Method section.(page 11, line 203-204).

4. This is the single sentence in the Results which deals with intraocular pressure rise aspect in the study. Intraocular pressure monitoring is standard care before and after intravitreal injection. Why do authors chose to report and combine intraocular pressure data along side systemic thromboembolic adverse events as both are completely unconnected. 

Reply: Thanks for your valuable comments.

We added additional information about glaucoma issue in the Discussion section(page 21, line 321-344).

Patients with chronic retinal diseases require multiple injections; therefore, the risks of systemic and ocular adverse events constitute a concern.The major systemic adverse events associated with the injection of anti-VEGF agents are arterial thromboembolic events(ATEs) , and the risks of ATEs between the two drugs are concerned due to many patients receiving intravitreal injection are high risks for ATEs , such as old aged patients and DM patients. In addition, frequently reported ocular adverse effects with anti-VEGF intravitreal injection include the cataract, glaucoma, and conjunctival hemorrhage.1,2 However, the increased risks of glaucoma are concerning due to the irreversible optic neuropathy that may result.Therefore, we chose glaucoma as an indicator of these ocular adverse events.

Because the injection procedure itself does not differ between two drugs, the medication itself may play a role in the systemic and ocular adverse events. Aflibercept had a longer half-life than ranibizumab, and inflammation appears more common in aflibercept than in ranibizumab.3 The inflammation could affect aqueous humor production or outflow pathways, such as the trabecular meshwork.4Previous studies had demonstrated that elevated inflammatory cytokines in POAG patients,5,6and chronic inflammation plays an important role in glaucoma mechanism.7Inflammation also plays a critical role in all stages of atherosclerotic plaques formation, leading to the acute coronary and cerebrovascular syndromes.8We discussed the issue about inflammation of these drugs in the revised manuscript(page 20, line 311-312, and page22, line 331-333).Addtional studies will be need to evaluate the systemic and intraocular inflammation reaction between these two drugs, and the link between the inflammation and the risks of those systemic and ocular adverse events.

References:

1. Holz FG, Figueroa MS, Bandello F, et al. RANIBIZUMAB TREATMENT IN TREATMENT-NAIVE NEOVASCULAR AGE-RELATED MACULAR DEGENERATION: Results From LUMINOUS, a Global Real-World Study. Retina. 2020;40(9):1673-1685. doi:10.1097/IAE.0000000000002670

2. Scott IU, VanVeldhuisen PC, Ip MS, Blodi BA, Oden NL, Awh CC, Kunimoto DY, Marcus DM, Wroblewski JJ, King J; SCORE2 Investigator Group. Effect of Bevacizumab vs Aflibercept on Visual Acuity Among Patients With Macular Edema Due to Central Retinal Vein Occlusion: The SCORE2 Randomized Clinical Trial. JAMA. 2017 May 23;317(20):2072-2087. doi: 10.1001/jama.2017.4568. PMID: 28492910; PMCID: PMC5710547.

3. Souied EH, Dugel PU, Ferreira A, Hashmonay R, Lu J, Kelly SP. Severe Ocular Inflammation Following Ranibizumab or Aflibercept Injections for Age-Related Macular Degeneration: A Retrospective Claims Database Analysis. Ophthalmic Epidemiol. 2016;23(2):71-79. doi:10.3109/09286586.2015.1090004

4. Ramsey, D.J., McCullum, J.C., Steinberger, E.E. et al. Intraocular pressure decreases in eyes with glaucoma-related diagnoses after conversion to aflibercept for treatment-resistant age-related macular degeneration. Eye (2021). https://doi.org/10.1038/s41433-021-01729-1

5. Ten Berge JC, Fazil Z, van den Born I, et al. Intraocular cytokine profile and autoimmune reactions in retinitis pigmentosa, age-related macular degeneration, glaucoma and cataract. Acta Ophthalmol. 2019;97(2):185-192. doi:10.1111/aos.13899

6. Burgos-Blasco B, Vidal-Villegas B, Saenz-Frances F, Morales-Fernandez L, Perucho-Gonzalez L, Garcia-Feijoo J, Martinez-de-la-Casa JM. Tear and aqueous humour cytokine profile in primary open-angle glaucoma. Acta Ophthalmol. 2020 Sep;98(6):e768-e772. doi: 10.1111/aos.14374. Epub 2020 Feb 11. PMID: 32043817.

7. Baudouin C, Kolko M, Melik-Parsadaniantz S, Messmer EM. Inflammation in Glaucoma: From the back to the front of the eye, and beyond. Prog Retin Eye Res. 2021 Jul;83:100916. doi: 10.1016/j.preteyeres.2020.100916. Epub 2020 Oct 17. PMID: 33075485.

8. Arnott C, Punnia-Moorthy G, Tan J, Sadeghipour S, Bursill C, Patel S. The Vascular Endothelial Growth Factor Inhibitors Ranibizumab and Aflibercept Markedly Increase Expression of Atherosclerosis-Associated Inflammatory Mediators on Vascular Endothelial Cells. PLoS One. 2016;11(3):e0150688. Published 2016 Mar 9. doi:10.1371/journal.pone.0150688

Minor comments.

9. In the Methods: “Because patients data are deidenetifed in the NHIRD before release to researchers, the requirement for informed consent under the full review process of the TMU-JIRB was waive.” Does the data storage policy include patients’ consent permitting use of their deidentifed data for research purpose? ‘Additionally, the study used the National Death Registry under the regulation of Health and Welfare Data Science Center, Ministry of Health and Welfare, Executive Yuan, Taiwan.’ The use of this data in the study needs to be elaborated. 

Response: Thanks for your comments. Because the study involved data derived from human subjects, patients’ privacy and data confidentiality are major concern. To protect the privacy of patients, patient data in the NHIRD were deidentified before they are released to researchers. The need for informed consent was waived because it was impossible to obtain from patients. According to the data agreement regulations, the use of the data must be for research purposes only. All applications are to be reviewed by peer experts to ensure rational data use. In addition, researchers must follow Taiwan’s Computer-Processed Personal Data Protection Law and related regulations and sign an agreement declaring that no attempt will be made to retrieve information that would potentially violate the privacy of patients or health-care providers.1 The NHIRD encrypts personal patient information to protect privacy and provides researchers with anonymous identification numbers associated with relevant claims information1, including sex, date of birth, medical services received, and prescriptions received. Therefore,patient consent is not required to access the NHIRD, and as such, the Joint Institutional Review Board of Taipei Medical University issued a formal written waiver to obviate the need for informed consent. This study was approved as fulfilling those conditions required for exemption by the Institutional Review Board of Taipei Medical University (TMU-JIRB No.202005104).We have added these information to the revised manuscript(page 7, line 125-130).

Moreover, the two major databases used in the study, namely the NHIRD and the National Death Registry, can be linked through a unique deidentifier. To protect personal information, researchers are required to perform their analyses on the servers provided by the Health and Welfare Data Center (HWDC), Ministry of Health and Welfare (MOHW), Taiwan. According to this regulation, no individual-level data can be taken out, and all results must be reviewed by data custodians to prevent any disclosure of patient identity.We have added additional relevant information to the revised manuscript(page 8, line145-153).

References

1. Hsieh CY, Su CC, Shao SC, et al. Taiwan's National Health Insurance Research Database: past and future. Clin Epidemiol. 2019;11:349-358. Published 2019 May 3. doi:10.2147/CLEP.S196293

.

10. In Discussion section: “Besides, the Singpore study was performed in a single cener; therefore, the data did not represent the national population”. Do authors really think that this criticism was valid in view of small size and population of city state Singapore. 

Reply: Thank you for your suggestions.We agreed that the criticism is inapporiate, and have deleted the sentence (page 18 , line 275). 

11. In Discussion section: “The present study showed the risk of ATEs in DME patients did not significantly differ between the IVA and IVR groups, but we did not specifically compare the risks of ATEs in DME patients with a history of such events.” The second part of this sentence is unclear.

Reply: Thank you for your suggestion. We revised the sentence “The present study showed the risk of ATEs in DME patients did not significantly differ between the IVA and IVR groups, but we did not specifically compare the risks of ATEs in DME patients with a history of APTC events.” (page 20 , line 300-301).

12. The section of Discussion dealing with “Risk of Glaucoma” appears nondescript and not clearly written. It does not drive home the intended message. 

Reply: Thanks for your comments, and we revised the Discussion section “Risk of Glaucoma” (page 21-22 , line 321-344).

.

---

## [Decision Letter · Decision Letter 4]

28 Jan 2022

PONE-D-20-25455R4Comparison of risks of Arterial Thromboembolic Events and Glaucoma with Ranibizumab and Aflibercept Intravitreous Injection: A Nationwide Population‐Based Cohort StudyPLOS ONE

Dear Dr. Lin,

Thank you for submitting your manuscript to PLOS ONE. After careful consideration, we feel that it has merit but does not fully meet PLOS ONE’s publication criteria as it currently stands. Therefore, we invite you to submit a revised version of the manuscript that addresses the points raised during the review process.

Specifically, please address the remaining concerns from reviewer 4.

We look forward to receiving your revised manuscript.

Kind regards,

Jianhong Zhou

Associate Editor

PLOS ONE

Journal Requirements:

Reviewers' comments:

Reviewer's Responses to Questions

**Comments to the Author**

1. If the authors have adequately addressed your comments raised in a previous round of review and you feel that this manuscript is now acceptable for publication, you may indicate that here to bypass the “Comments to the Author” section, enter your conflict of interest statement in the “Confidential to Editor” section, and submit your "Accept" recommendation.

Reviewer #3: All comments have been addressed

Reviewer #4: (No Response)

2. Is the manuscript technically sound, and do the data support the conclusions?

Reviewer #3: Yes

Reviewer #4: Partly

3. Has the statistical analysis been performed appropriately and rigorously? 

Reviewer #3: Yes

Reviewer #4: Yes

4. Have the authors made all data underlying the findings in their manuscript fully available?

Reviewer #3: Yes

Reviewer #4: No

5. Is the manuscript presented in an intelligible fashion and written in standard English?

Reviewer #3: Yes

Reviewer #4: Yes

6. Review Comments to the Author

Reviewer #3: Authors have done a commendable job with the study. All the comments have been adequately answered. This study is scientifically sound. I have no further comments.

Reviewer #4: Definition of glaucoma used to identify cases needs to be provided in the Methods section. It needs to be discretely mentioned.

7. PLOS authors have the option to publish the peer review history of their article (what does this mean?). If published, this will include your full peer review and any attached files.

Reviewer #3: No

Reviewer #4: **Yes: **Kumar Saurabh

---

## [Author Response · Author response to Decision Letter 4]

28 Feb 2022

Reviewers' comments:

1. Have the authors made all data underlying the findings in their manuscript fully available? The PLOS Data policy requires authors to make all data underlying the findings described in their manuscript fully available without restriction, with rare exception (please refer to the Data Availability Statement in the manuscript PDF file). The data should be provided as part of the manuscript or its supporting information, or deposited to a public repository. For example, in addition to summary statistics, the data points behind means, medians and variance measures should be available. If there are restrictions on publicly sharing data—e.g. participant privacy or use of data from a third party—those must be specified.

Reply: Thanks for your comments. We have added the Data Availability statement in the revised manuscript (page 24, line 377-381).

2. Reviewer 4: Definition of glaucoma used to identify cases needs to be provided in the Methods section. It needs to be discretely mentioned.

Reply: Thanks the valuable comment. We have added more description of the definition of glaucoma cases in the revised manuscript (page 11, line 201-208).

---

## [Decision Letter · Decision Letter 5]

4 Apr 2022

Comparison of risks of Arterial Thromboembolic Events and Glaucoma with Ranibizumab and Aflibercept Intravitreous Injection: A Nationwide Population‐Based Cohort Study

PONE-D-20-25455R5

Dear Dr. Lin,

We’re pleased to inform you that your manuscript has been judged scientifically suitable for publication and will be formally accepted for publication once it meets all outstanding technical requirements.

Kind regards,

James Mockridge

Staff Editor

PLOS ONE

Reviewers' comments:

Reviewer's Responses to Questions

**Comments to the Author**

1. If the authors have adequately addressed your comments raised in a previous round of review and you feel that this manuscript is now acceptable for publication, you may indicate that here to bypass the “Comments to the Author” section, enter your conflict of interest statement in the “Confidential to Editor” section, and submit your "Accept" recommendation.

Reviewer #3: All comments have been addressed

Reviewer #4: All comments have been addressed

2. Is the manuscript technically sound, and do the data support the conclusions?

Reviewer #3: Yes

Reviewer #4: Yes

3. Has the statistical analysis been performed appropriately and rigorously? 

Reviewer #3: Yes

Reviewer #4: Yes

4. Have the authors made all data underlying the findings in their manuscript fully available?

Reviewer #3: Yes

Reviewer #4: Yes

5. Is the manuscript presented in an intelligible fashion and written in standard English?

Reviewer #3: Yes

Reviewer #4: Yes

6. Review Comments to the Author

Reviewer #3: Dear author, I thank you for revising the manuscript. Even though there may be ethnical variations in the rates of adverse events and glaucoma in eyes undergoing intravitreal injections, the study makes a reasonable claim that aflibercept can be accepted as a reasonable alternative if an intravitreal injection is needed. I congratulate you for your efforts.

Reviewer #4: Though authors have provided definition of glaucoma, it would still be better if the same was written in the text itself.

7. PLOS authors have the option to publish the peer review history of their article (what does this mean?). If published, this will include your full peer review and any attached files.

Reviewer #3: No

Reviewer #4: No

---

## [Editor Report · Acceptance letter]

8 Apr 2022

PONE-D-20-25455R5 

Comparison of risks of arterial thromboembolic events and glaucoma with ranibizumab and aflibercept intravitreous injection: A nationwide population‐based cohort study 

Dear Dr. Lin:

I'm pleased to inform you that your manuscript has been deemed suitable for publication in PLOS ONE. Congratulations! Your manuscript is now with our production department. 

Kind regards, 

on behalf of

Dr James Mockridge 

Staff Editor

PLOS ONE